# Aircraft Evaluation of MODIS Cloud Drop Number Concentration Retrievals

Scarlet R. Passer[1], Mikael K. Witte[2], and Patrick Y. Chuang[1]

[1]Earth and Planetary Sciences, University of California Santa Cruz, Santa Cruz, CA, USA
[2]Department of Meteorology, Naval Postgraduate School, Monterey, CA, USA

**Correspondence:** Patrick Y. Chuang (pchuang@ucsc.edu)

**Abstract.** Cloud drop number concentration ($N_d$) can be retrieved through passive satellite observation. These retrievals are useful due to their wide spatial and temporal coverage. However, the accuracy of the retrieved values is not well understood. In this study, we seek to understand why the retrievals agree or disagree with in situ measurements by examining the various cloud properties that underlie the retrievals. To do so, we compare satellite $N_d$ derived from the Moderate Resolution Imaging Spectroradiometer (MODIS) instrument with in situ aircraft measurements using a phase Doppler interferometer onboard three flight campaigns sampling marine stratocumulus clouds. Intercomparison of $N_d$ values shows that the discrepancy between retrieved and in situ $N_d$ can be $\pm 50\%$ or more. In the mean, there is evidence of an overestimation bias by MODIS retrievals, although the sample size is insufficient for statistical certainty. We find that MODIS $N_d$ is best interpreted as representative of the mid-cloud region, as there is almost always a greater discrepancy from in situ values near cloud top and cloud base. We also find evidence of cases where $N_d$ is accurately retrieved, but effective radius is not, presumably due to offsetting errors in other retrieval parameters. Vertical profiles of extinction coefficient $\beta$, liquid water content $L$, and effective radius $r_e$ measured during sawtooth-pattern flight legs through cloud top are also compared to implicit MODIS retrieval profiles. For the two cases with $N_d$ agreement, all profiles match well. For the six cases with significant disagreement, there is no consistent underlying cause. The discrepancy originates from either: (a) discrepancy in the $r_e$ profile, (b) discrepancy in the $\beta$ and $L$ profiles, or (c) discrepancy in both.

## 1 Introduction

Cloud drop number concentration ($N_d$) is a fundamental property of clouds. This property is relevant to numerous effects that clouds have on weather and climate. For example, clouds with higher $N_d$ will have a larger albedo (assuming all else equal), which in turn will lead to a larger solar reflectance (Twomey, 1977). The ability of a warm (i.e., ice-free) cloud to form precipitation is also dependent on $N_d$. A cloud with very high $N_d$ will not produce appreciable rainfall rates since the drops are unable to grow to the required sizes to efficiently sediment, while at the other extreme of very low $N_d$, warm clouds produce precipitation very effectively (Sorooshian et al., 2009). Understanding and evaluating aerosol-cloud interactions frequently involves $N_d$, since activation of particulate matter is one of the strongest connections between aerosol and clouds (Bellouin et al., 2020; Quaas et al., 2020; Gordon et al., 2023).

Passive satellite observation is one of the primary methods that we have to measure and understand the climatology of $N_d$ at large spatial and temporal scales (e.g., Bennartz, 2007; McCoy et al., 2018, 2020; Christensen et al., 2022). The Moderate Resolution Imaging Spectroradiometer (MODIS) instrument onboard the Terra and Aqua satellites is one important source of satellite observations of $N_d$. A review paper by Grosvenor et al. (2018) comprehensively surveys our understanding of $N_d$ derived from remote sensing, so we focus this Introduction only on aspects that are directly relevant for this study. A

theoretical analysis by Bennartz (2007) suggests that for cloud fractions over 80% and LWP>30 g/m$^2$, i.e., conditions relevant to the stratocumulus (Sc) clouds observed in this study, MODIS $N_d$ retrievals should exhibit relative errors < 80%, a value which Grosvenor et al. (2018) is in agreement with. These errors are mainly due to uncertainties in retrieving $r_e$, LWP, cloud optical depth $\tau_c$, and cloud fraction.

      There are a handful of past studies that assess the agreement between in situ aircraft measurements and satellite retrievals

of $N_d$. Painemal and Zuidema (2011) use data from the National Center for Atmospheric Research (NCAR) C-130 aircraft measuring Sc in the SE Pacific during VOCALS. They find good agreement in the mean for the 19 cases that they examined, while using the assumption that the liquid water profile is adiabatic. Two of the 19 cases had large discrepancies (30 to 50% discrepancy) that they attribute to sub-adiabaticity. Their cases span a wide range, from <50 cm$^{-3}$ to >300 cm$^{-3}$. They also find that the assumption that $N_d$ is constant with height was valid for their cases. However, Grosvenor et al. (2018) point out

that their strong agreement is not necessarily because the underlying measurements are accurate, but due to an overestimate of the effective radius almost exactly offsetting an overestimate of the cloud adiabaticity. Min et al. (2012) also analyze data from VOCALS, but combine NCAR C-130 and US Department of Energy (DOE) G-1 aircraft data, and find that, without accounting for adiabaticity, MODIS overestimates $N_d$ relative to aircraft measurements, with a mean bias of 25%. If measured adiabaticity is used, then no statistically significant bias is detected, with most of their 17 cases appearing to agree to within

50% (two outliers are noted). A third study (Bennartz and Rausch, 2017) compared results from their updated algorithm to the same VOCALS results as Painemal and Zuidema (2011). They find a modest bias (less than 20 cm$^{-3}$) between aircraft and MODIS, and an average uncertainty of $\sim 35$ cm$^{-3}$. McCoy et al. (2018) utilize a much larger data set, but in order to do so, they greatly relaxed the requirements for co-location of aircraft and satellite observations in both space and time. They compare aircraft measurements with the satellite retrieval averaged over 3 days for the closest 3°× 3° area. They find average

agreement to be much poorer than these previous studies ($r^2$=0.46), but this may simply be a matter of not measuring the same cloud at the same time.

      Gryspeerdt et al. (2022) compared many aircraft field campaigns with satellite retrieved $N_d$, although only a subset of these are focused on stratiform clouds. Their data set is a mixture of different instruments and different aircraft platforms. If we examine their results only from Sc projects, they find $r^2$ values between 0.5 and 0.75. Restricting the data using different

filters can somewhat increase $r^2$, though the range for Sc-only campaigns remains about the same. The best-fit slope of the in situ vs. MODIS data for Sc-only campaigns is, with one exception (discussed next), close to 1, suggesting no mean bias. Variability of the data is high, however, with many measurements disagreeing by a factor of 2 or more. The one project that is the exception, i.e. exhibits significant bias between in situ and satellite derived $N_d$ is, surprisingly, from the analysis of NCAR C-130 data from VOCALS. The slope is very noticeably different from 1, with MODIS exhibiting larger values relative to

aircraft. This result appears to disagree with the three other studies that focus on VOCALS (described above). The source of this disagreement is unclear.

Broadly speaking, these comparisons, with some exceptions, paint a rather optimistic picture for $N_d$ retrievals. Most of these studies suggest that MODIS retrievals are better than the theoretical 80% relative uncertainty proposed by Bennartz (2007), albeit these are not all the same retrievals (many of the studies filter the data in various ways to assess which conditions are most favorable for an accurate retrieval). However, given that many of these studies focus on the VOCALS field campaign and utilize in situ cloud probes with the same operating principle (with Gryspeerdt et al., 2022, being the exception to both), it would be informative to broaden the type of cloud and instrument used for evaluating satellite $N_d$ retrievals. There are reasons to believe that the in situ measurements onboard the C-130 from the VOCALS campaign may have biases. Past studies (e.g. Painemal and Zuidema, 2011) attribute biases between in situ and MODIS-retrieved effective radius during VOCALS to issues with the satellite retrievals (e.g., due to three-dimensional radiative effects or unresolved horizontal heterogeneity). However, Witte et al. (2018), using aircraft data from a phase Doppler interferometer (PDI) during three different Sc field campaigns, found no bias between in situ and MODIS $r_e$. They find that the bias when using the older cloud probes is correlated with the breadth of the drop size distribution, which they attribute to difficulties that such probes have measuring larger drops. However, this issue may affect estimation of $r_e$ differently than for $N_d$, so it is unclear what the consequences are, if any, of such instrument limitations.

The focus of this study is to compare MODIS retrievals against in situ measurements from three Sc-focused field campaigns using PDI data from the Twin Otter in order to evaluate the accuracy of the satellite $N_d$ product. These are the same data sets as used in Witte et al. (2018). Importantly, we will also evaluate the validity of the assumptions underlying the retrievals through analysis of vertical profiles of cloud properties, which, to our knowledge, previous studies have not examined in detail. When we find good agreement, is it because the underlying properties are also in agreement, or are there cases of canceling errors? When we find poor agreement, is there one consistent reason for it, or is there a diversity of reasons?

## 1.1 Satellite $N_d$ retrieval

The most common retrieval of number concentration utilizes the following equation (Grosvenor et al., 2018):

$$N_d = \frac{\sqrt{5}}{2\pi k} \left( \frac{f_{ad} c_w \tau_c}{Q_{ext} \rho_w r_e^5} \right)^{1/2} \qquad (1)$$

Cloud optical depth $\tau_c$ and cloud top effective radius $r_e$ are the two (mostly) independently retrieved quantities used in the $N_d$ retrieval where $r_e$ is defined as:

$$r_e = \frac{\int_0^\infty r^3 n(r) dr}{\int_0^\infty r^2 n(r) dr} \qquad (2)$$

where $n(r)$ is the drop size distribution as a function of drop radius $r$. Retrieved $N_d$ is more sensitive to the same relative uncertainty in retrieved $r_e$ than $\tau_c$ due to the difference in the magnitude of the exponents (5/2 versus 1/2) in Eq. 1. Density

of water $\rho_\mathrm{w}$ is known, and the remaining variables (adiabatic fraction $f_\mathrm{ad}$, water content lapse rate $c_\mathrm{w}$, a constant that relates the mean-volume and effective radii $k$, and the extinction efficiency factor $Q_\mathrm{ext}$) are considered fixed as described below. To produce an estimate of $r_\mathrm{e}$ from aircraft suitable for comparison with that from MODIS, a weighting function is used to weight the impact of cloud vertical structure on the aircraft-derived variables (Platnick, 2000):

$$W(\tau_\mathrm{c}) = a\tau_\mathrm{c}^b exp\left(-\tau_\mathrm{c}\left(\frac{1}{\mu} + \frac{1}{\mu_0}\right)\right) \tag{3}$$

Here, $b = 2$, $a$ is a normalization constant, and $\mu$ and $\mu_0$ are the cosine of the sensor and solar zenith angles, respectively. The weighting function describes, as a function of cloud optical depth, how much influence the measurement from a given region of a cloud has on satellite-derived variables. This function peaks within a few tens of meters of cloud top (Platnick, 2000; Witte et al., 2018), and therefore the effective radius reflects values in this cloud top region.

Cloud optical depth $\tau_\mathrm{c}$ is defined as the vertical integral of the extinction coefficient $\beta(z)$:

$$\tau_\mathrm{c} = \int_{z_\mathrm{base}}^{z_\mathrm{top}} \beta(z)dz \tag{4}$$

where $z_\mathrm{top}$ and $z_\mathrm{base}$ are the altitude of cloud top and cloud base, respectively.

At any altitude, $\beta(z)$ is related to the cloud drop size distribution $n(r,z)$, for which we now include the dependence on altitude $z$:

$$\beta(z) = \int_0^\infty \pi Q_\mathrm{ext} n(r,z)r^2 dr \tag{5}$$

The remaining variables in the $N_\mathrm{d}$ retrieval (Eq. 1) are described as follows:

1. $N_\mathrm{d}$ is assumed to be a constant with respect to height in the cloud, i.e. $N_\mathrm{d}(z) = $ constant.

2. $k$ is defined as:

$$k = \left(\frac{r_\mathrm{v}}{r_\mathrm{e}}\right)^3 \tag{6}$$

where $r_\mathrm{v}$ is volume-mean drop radius. The MODIS retrieval assumes that $k = 0.8$. Previous studies suggest that $k$ is well-constrained in stratocumulus clouds, typically ranging between 0.7 and 0.9 (Miles et al., 2000; Lebsock and Witte, 2023).

3. $Q_\mathrm{ext}$ is the extinction efficiency (dimensionless), and represents the ratio between the extinction and geometric cross sections of a drop. It is a function of drop radius $r$, but because the radius of the drops of interest is usually much larger than the wavelengths of light used for the retrievals, it can be assumed $Q_\mathrm{ext} = 2$ (the limit for geometric optics) (Platnick, 2000).

4. $c_w$ is the adiabatic gradient of liquid water content $L$ with respect to height, and is a weak function of temperature and pressure. From Eq. 14 in Grosvenor et al. (2018), we compute a value of $c_w = 2.3 \times 10^{-6}$ kg/m$^4$. We assume $c_w$ to be constant vertically through a cloud, which should introduce an error that is less than 1% since the stratocumulus observed in this study are quite geometrically thin ($< 500$ m) (Grosvenor et al., 2018).

5. $f_{ad}$ is defined as the fraction of cloud liquid water content relative to its adiabatic value at a given height above cloud base. The MODIS retrieval assumes $f_{ad} = 0.6$. Combining the definitions of $c_w$ and $f_{ad}$ yields the profile of liquid water content:

$$L(z) = f_{ad} c_w (z - z_{base}) \tag{7}$$

### 1.1.1 Implicit retrieval profiles

The MODIS retrieval implicitly assumes specific vertical profiles of $r_e$, $\beta$, and $L$. These profiles, along with cloud base height, can be derived as follows:

1. The cloud top liquid water content is computed as:

$$L(z_{top}) = \frac{4}{3} \pi \rho_w \cdot k r_e^3(z_{top}) \cdot N_d \tag{8}$$

2. The liquid water content profile $L(z)$ is defined above (Eq. 7).

3. Cloud base height $z_{base}$ is determined by the altitude where $L(z) = 0$. By re-arranging Eq. 7 and applying it at $z = z_{top}$, we get:

$$z_{base} = z_{top} - L(z_{top})/f_{ad} c_w \tag{9}$$

4. $r_e(z)$ is derived starting from the definition of $r_v$:

$$\frac{4}{3} \pi r_v^3(z) N_d \rho_w = \frac{4}{3} \pi \left[ k r_e^3(z) \right] N_d \rho_w = L(z)$$

which can be re-arranged in terms of $r_e$:

$$r_e(z) = \left[ \frac{3}{4 \pi \rho_w k} \frac{L(z)}{N_d} \right]^{1/3} \tag{10}$$

5. $\beta(z)$ is derived by substituting the definitions of $r_e$ and $L$ into Eq. 5, which yields (Grosvenor et al., 2018):

$$\beta(z) = \frac{3}{4} \frac{Q_{ext}}{\rho_w} \frac{L(z)}{r_e(z)} \tag{11}$$

In order to better identify the source of any discrepancies between MODIS and in situ $N_d$, the vertical profiles $r_e(z)$, $\beta(z)$, and $L(z)$ that are inherent in the MODIS algorithm for estimating $N_d$ will be compared to in situ observations of the same quantities, along with cloud base height $z_{base}$. There are a number of potential sources of uncertainty, including the MODIS retrievals of $r_e$ and $\tau_c$, as well as the validity of the above assumptions.

## 2   Methods

### 2.1   In Situ Observations of $N_d$

#### 2.1.1   Aircraft Observations

In situ data was acquired during three different flight campaigns that sampled marine stratocumulus: the Marine Stratus/Stratocumulus Experiment (MASE;  Lu et al., 2007), the Physics of Stratocumulus Top experiment (POST;  Carman et al., 2012; Gerber et al., 2013), and the Variability of American Monsoon Systems Ocean-Cloud-Atmosphere-Land Study (VOCALS; Mechoso et al., 2014; Zheng et al., 2011). All three of these campaigns used a phase Doppler interferometer (PDI) onboard the CIRPAS Twin Otter (TO) aircraft to derive cloud microphysical properties. See Chuang et al. (2008) for details about the PDI measurement method and data processing.

The MASE campaign was flown in the NE Pacific near Monterey, California during July 2005. VOCALS was centered off the coast of Chile in the SE Pacific and was flown during October to November of 2008. Flights during the MASE and VOCALS campaigns utilized level legs (i.e., flight segments flown at constant altitude and heading for a sustained period, usually 10 min for the flights analyzed in this work) which sampled from below cloud base to near cloud top. POST was flown slightly farther offshore in a similar location as MASE, during July and August of 2008. From the POST campaign we analyze flight legs that were flown in a sawtooth pattern, flying repeatedly up and down between $\sim 100$ m below cloud top to $\sim 100$ m above. Overall, we analyze four flight days from MASE, ten from VOCALS, and eight from the POST campaign, as these flights coincide with a MODIS overpass. More details on matching the aircraft flights with MODIS overpasses can be found in Witte et al. (2018).

The PDI measurements from these three field campaigns have been used to analyze the retrieval of cloud effective radius from MODIS. Previous studies (Painemal and Zuidema, 2011; Min et al., 2012; Noble and Hudson, 2015) have suggested that MODIS retrievals of $r_e$ are biased high by 2 to 5 µm relative to in situ measurements, but using the same data set as in this study, Witte et al. (2018) found no such bias, with MODIS and in situ measurements agreeing within 0.7 µm in the mean. They attribute observed bias to issues with the in situ aircraft instrumentation used in previous studies. This suggests that, in the mean, there is no bias in retrievals of $N_d$ due to a bias in retrieved $r_e$, an assertion that we will evaluate as part of our analysis.

#### 2.1.2   In Situ $N_d$ and $r_e$ Calculations

To estimate cloud drop number concentration from PDI, in-cloud sampling legs are analyzed for each flight. To be consistent with MODIS, data from near cloud top are used to derive $r_e$. In contrast, because the satellite retrieval assumes $N_d$ is constant

throughout the cloud, mid-cloud data are used to derive number concentration, as this location is more representative of the mean cloud $N_d$ value relative to cloud top values (as will be shown below).

    For the VOCALS and MASE campaigns, we analyze mid-cloud and cloud top level legs. However, the POST campaign primarily used a sawtooth flight pattern. Therefore, we define cloud top as the altitude where liquid water content crosses a threshold of $L = 0.05$ g/m$^3$, a commonly used threshold in the airborne science community. Effective radius is calculated from

within 10 m of this cloud top. We use the range between 60 m to 90 m below cloud top as an analog to level mid-cloud legs to calculate a representative value of $N_d$ during POST. While this altitude range does not correspond to "mid-cloud" in the sense of cloud geometric thickness, this region is typically far enough from cloud top to avoid the impacts of entrainment mixing.

    For ease in comparing MODIS implicit cloud profile estimations to observation, we create a shifted altitude ($z_{\text{shift}}$) for POST which is defined by $z_{\text{shift}} = 0$ at cloud top, i.e. $z_{\text{shift}} = z - z_{\text{top}}$. To perform this coordinate transformation, we determine the

altitude of cloud top ($z_{\text{top}}$) for each individual ascent or descent (or "leg") of the sawtooth flight path (using the threshold $L = 0.05$ g/m$^3$). All in situ measurements for each leg are referenced to $z_{\text{shift}}$ for that leg. More details about the altitude shifting process can be found in Carman et al. (2012).

    For the purpose of making a more like-to-like comparison between PDI and MODIS, number concentration and effective radius measured along their respective legs are averaged over 1 km (20 s) intervals to match MODIS spatial resolution. The

mean and variability of $N_d$ and $r_e$ for each 10 min (or $\approx$30 km at a mean true airspeed of 55 m/s) flight leg are calculated from these 1 km average values, which is the uncertainty that we show in the results (below). This variability does not reflect any uncertainty due to differences in the spatial domain sampled, as well as any temporal differences in the sampling, which are difficult to assess.

    The main source of instrumental uncertainty is the uncertainty in the instrument view volume. The view volume, in units of

volume of air sampled per second, is the product of three values. The first is the probe "length", which is calculated for each flight from the collected data itself (using the method from Chuang et al., 2008), so day-to-day variation should be accounted to well within 5%. The second value is the aircraft true air speed, which is known quite accurately, almost certainly to within 5%. The third value is the probe "width", which is fixed by the optical hardware. Recent in-depth laboratory examination of this subject (Leandro, 2023) suggests that this width may be smaller than the theoretical value for very small drops, which

would lead to an under-estimation of $N_d$, but mostly affect larger values of $N_d$ (which tend to exhibit smaller drop sizes). We estimate that this bias could be as large as 10% for $N_d > 400$ cm$^{-3}$ and decreases to less than 1% for $N_d < 100$ cm$^{-3}$. Counting uncertainty is unlikely to be significant because many thousands of drops are observed for any given data point, so the Poisson standard deviation of $\sqrt{n}$ would suggest an uncertainty of $< 1\%$. These instrumental uncertainties, when combined, produce an uncertainty of less than 20%, which is almost always smaller than the observed spatiotemporal variability, which

is why we report the latter.

### 2.1.3   Profile Calculations

The $N_d$ retrieval combines measurements of $\tau_c$ and cloud top $r_e$ with assumptions about the cloud vertical structure (see Section 1.1). Therefore, specific vertical profiles of effective radius, liquid water content, and extinction coefficient are implicitly

assumed. Due to the sawtooth sampling strategy during the POST campaign, we can compare these assumed profiles with observations. We also evaluate the assumptions that $k$ and $f_{ad}$ are constant.

Profiles of $L$, $r_e$, and $\beta$ are derived directly from cloud drop size distributions measured by the PDI and binned over 5 m increments within $z_{shift}$ space. Consistent with the satellite retrieval (Eq. 1), we also assume $Q_{ext} = 2$ when calculating $\beta$. To determine the adiabatic liquid water content $L_{ad}(z)$, first the mean altitude for near-cloud base is identified using a threshold liquid water content of $L = 0.1$ g/m$^3$. Next, $c_w$ is used to extrapolate downward to $L = 0$ g/m$^3$ associated with the true cloud base $z_{base}$.

## 2.2 Satellite retrieval details and sampling methodology

We utilize MODIS collection 6.1 level 2 retrievals of $r_e$ and $\tau_c$ using the 2.1 μm band ("Cloud_Effective_Radius" and "Cloud_Optical_Thickness" products, respectively) from both Aqua and Terra, consistent with the approach of Witte et al. (2018). Over the small sample size considered here, we find no evidence of systematic differences between satellites. Number concentration $N_d$ is then calculated from $r_e$ and $\tau_c$ using Eq. 1. A full description of the MODIS sampling methodology is given in Witte et al. (2018). Briefly, we analyze flight legs that occurred within 90 min of MODIS overpasses that covered the flight sampling area (most retrievals were within 30 min of their associated aircraft leg). We then select a 25 km$^2$ ($5 \times 5$ pixels) area centered on the mean coordinates of the aircraft leg to compare with in situ measurements.

## 2.3 Uncertainty calculations

Uncertainty of any variable $x$ is expressed as the 95% margin of error:

$$error = \frac{\sigma_x}{\sqrt{n_x}} z(0.95) \tag{12}$$

where $\sigma_x$ is the standard deviation of $x$, $n_x$ represents the total number of measurements, and the quantity $z(0.95)$ is the z-score for the 95% confidence interval and has a value of 1.96. We choose margin of error versus standard error to reflect uncertainty in unsampled spatial variability since the aircraft samples a narrow transect through a $1 \times 1$ km$^2$ box while MODIS senses photons arriving from the entire area.

The MODIS uncertainty is calculated from 1 km measurements over a 25 km$^2$ area. PDI uncertainty is calculated from downsampled 1 km measurements over the relevant time interval of a given flight leg. Both the MODIS and PDI uncertainties are a reflection of spatial variability at 1 km. This measurement does not reflect instrumental uncertainty or error in assumptions.

## 3 Results

We first analyze $N_d$ and $r_e$ for all three campaigns. After that, we capitalize on the sawtooth legs from the POST flights to compare profiles of $k$, $f_{ad}$, $r_e$, $\beta$, and $L$.

## 3.1 Comparison between MODIS and PDI $N_d$

Fig. 1 compares MODIS and PDI $N_d$ for all three flight campaigns. Though the majority of flights agree to within 50% (accounting for variability), there are several cases that exceed this range. Of those cases, most are overestimates by MODIS, with the one exception being VOCALS day 2008/11/01 (teal diamond in Fig. 1). At a population level, linear regression yields a slope of $1.1 \pm 0.14$ (95% confidence interval). Although the one-to-one line is included in this confidence interval, there is a suggestion of an overestimation bias. However, because of the limited sample size it is unclear if this is statistically significant.

There are several questions which arise from this result. First, when MODIS and PDI number concentration are in very good agreement (i.e., within sampling uncertainty of each other), is the satellite retrieved $N_d$ correct for the right underlying reasons or are there multiple errors which offset each other? Second, when MODIS disagrees with PDI, which MODIS retrieval products and/or assumptions are responsible for the discrepancy? To investigate these questions, we further analyze and compare the underlying variables in the MODIS $N_d$ retrieval.

## 3.2 Comparison between MODIS and PDI $r_e^{5/2}$

Effective radius is the most influential term in the $N_d$ calculation, and thus it's logical to speculate that good agreement between satellite and in situ $r_e$ values would manifest as good agreement between PDI and MODIS number concentration. Because MODIS $N_d \propto r_e^{-5/2}$ (Eq. 1), Fig. 2 compares MODIS and in situ $r_e^{5/2}$ values across the three campaigns.

Fig. 2 shows that there is a range of agreement between PDI and MODIS, with most of the days agreeing to within 25%. A linear regression of the data produces a slope of $1.0 \pm 0.07$ (95% confidence interval). This is a result consistent with Witte et al. (2018) who found no significant bias between MODIS and PDI measured $r_e$. Additionally, if $r_e$ were the determining factor in the accuracy of MODIS $N_d$ retrievals, it would be expected that the potential high bias of MODIS $N_d$ (Fig. 1) would correspond to a low bias in MODIS $r_e$. The data does not support this hypothesis, implying that the problem with the satellite estimation cannot solely be attributed to effective radius.

On an individual flight basis, comparing Fig. 2 and Fig. 1 shows that agreement in $N_d$ does not necessarily equate to agreement in $r_e^{5/2}$ and vice versa. In fact, there are four different scenarios which occur:

1. Agreement in $r_e^{5/2}$ with agreement in $N_d$.

2. Agreement in $r_e^{5/2}$ with disagreement in $N_d$.

3. Disagreement in $r_e^{5/2}$ with agreement in $N_d$.

4. Disagreement in $r_e^{5/2}$ with disagreement in $N_d$.

We illustrate one example of each case in Table 1. The finding that the $r_e$ retrieval does not govern agreement of MODIS $N_d$ leads us to further investigate the other variables in the MODIS $N_d$ calculation.

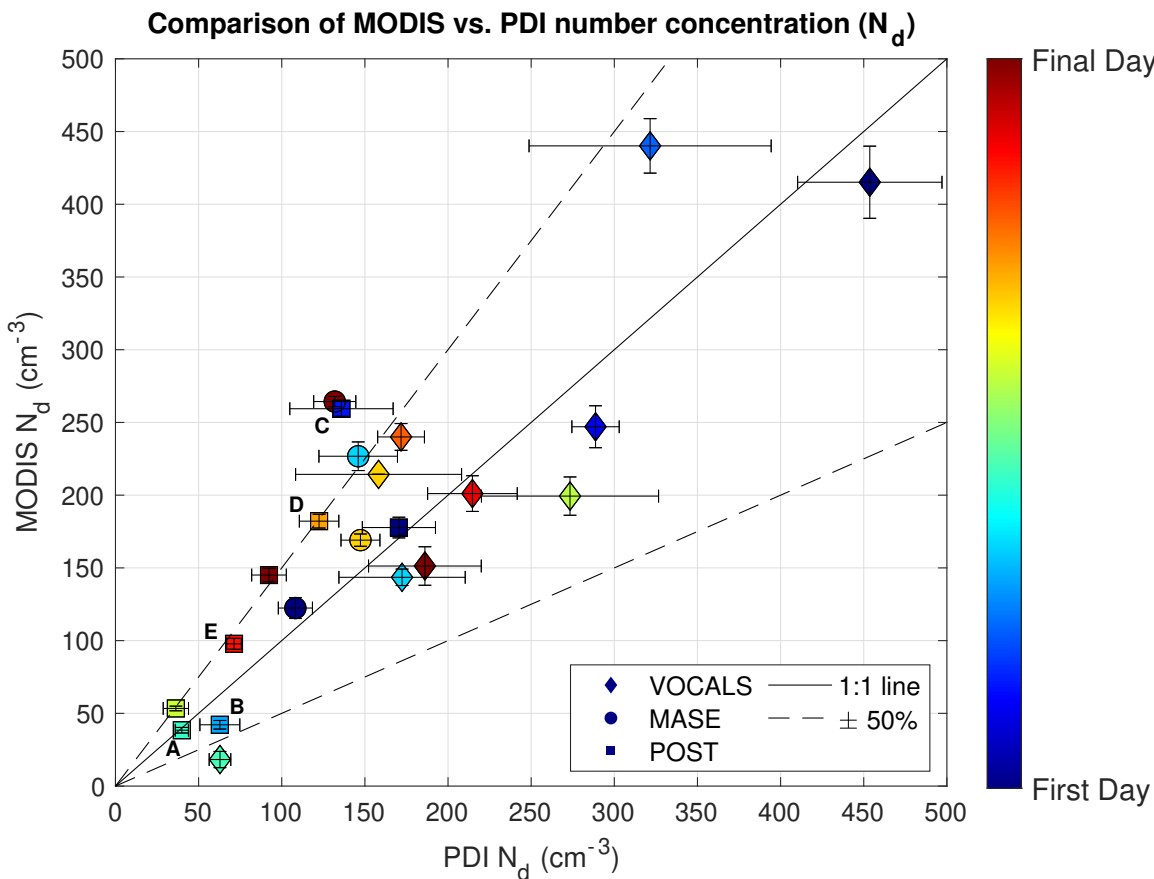

**Figure 1.** Comparison of MODIS number concentration vs PDI number concentration. PDI values are $N_d$ averaged at mid-cloud for all flight legs on that day, and the uncertainty bar represents $1\sigma$ variability of 1-km averaged values. Each flight campaign is marked by a symbol with color ranging from the first day of the campaign to the last. Several POST days are labeled A to E which represent specific cases discussed in Section 3.3. The MODIS $N_d$ value is a 1 km average of a $5\times5$ km$^2$ swath (with $1\sigma$ variability uncertainty bars) using Eq. 1 where effective radius and optical depth are taken from the satellite products. A linear fit of the data produces a slope of $1.1 \pm 0.14$.

| Good $N_d$ Agreement | | | | | |
|---|---|---|---|---|---|
| | Symbol | Campaign | Date | %Discrepancy $r_e$ | %Discrepancy $N_d$ |
| Good $r_e$ Agreement | ■ | POST | 2008/07/16 | $-6.8\%$ | $+4.1\%$ |
| Poor $r_e$ Agreement | ◆ | VOCALS | 2008/11/10 | $+37.0\%$ | $+4.0\%$ |
| Poor $N_d$ Agreement | | | | | |
| | Symbol | Campaign | Date | %Discrepancy $r_e$ | %Discrepancy $N_d$ |
| Good $r_e$ Agreement | ■ | POST | 2008/08/04 | $-0.04\%$ | $+33\%$ |
| Poor $r_e$ Agreement | ◆ | VOCALS | 2008/11/01 | $+17\%$ | $-250\%$ |

**Table 1.** Examples of each of the four possible combinations of $N_d$ and $r_e$ agreement with their associated campaign, day, discrepancy, and corresponding symbol from Figs. 1 and 2. The finding that at least one case has a large discrepancy (37%) in effective radius, with a small discrepancy (4%) in $N_d$ implies that there are compensating errors in other retrieval parameters. In other words, it is possible to retrieve the right $N_d$ for the wrong underlying reasons.

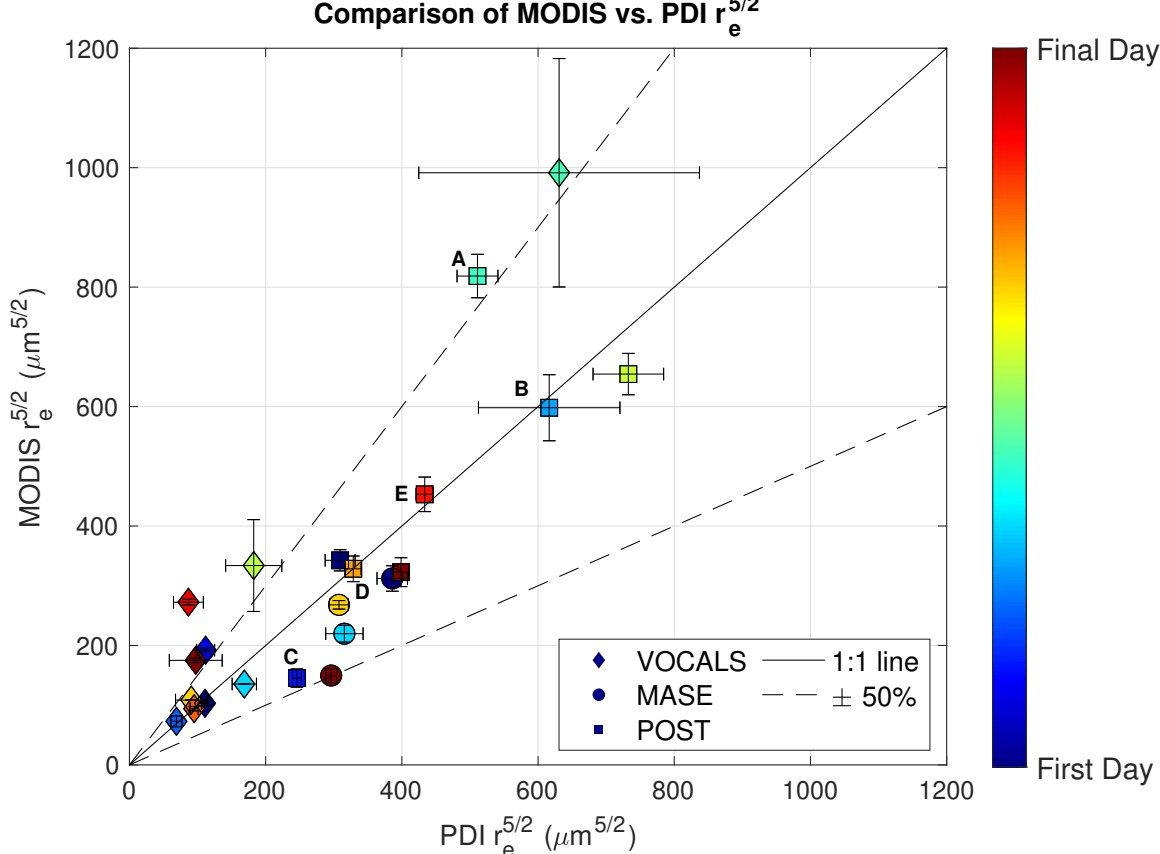

**Figure 2.** Comparison of $r_e^{5/2}$ between PDI measurements and the MODIS retrieved values. Satellite effective radius is the average of $5 \times 5$ km$^2$ swaths of the MODIS $r_e$ product. The PDI value is $r_e$ averaged for all cloud top flight legs on that day. A linear fit of the data produces a slope of $1.0 \pm 0.07$. Labels and uncertainty bars are the same as in Fig. 1.

### 3.3 Specific POST cases

The MODIS calculation for number concentration implicitly relies on vertical profiles of $L$, $r_e$, and $\beta$. These profiles combine MODIS measurements of $\tau_c$ and $r_e$ with assumptions of cloud vertical structure (see Section 1.1). We compare these profiles with those measured by aircraft during the POST campaign. The sawtooth flight plan of POST allows for detailed profile comparisons. Five POST cases are selected to illustrate the range of behaviors observed across all eight POST flights analyzed

in this manner. Fig. 3 illustrates the amount of data for four of the cases. Most of the data is concentrated within 150 m of cloud top. It is difficult to make statistically robust conclusions at altitudes below this region. Within these eight cases, we find two with excellent agreement between all observed and retrieved properties (one of which is explored in depth, Case A). The other six are cases where $N_d$ is not accurately retrieved. Among these cases, none represents an accurate retrieval of $N_d$

with significant errors in the underlying retrieval properties that compensate for each other, although such cases do exist (see Table 1).

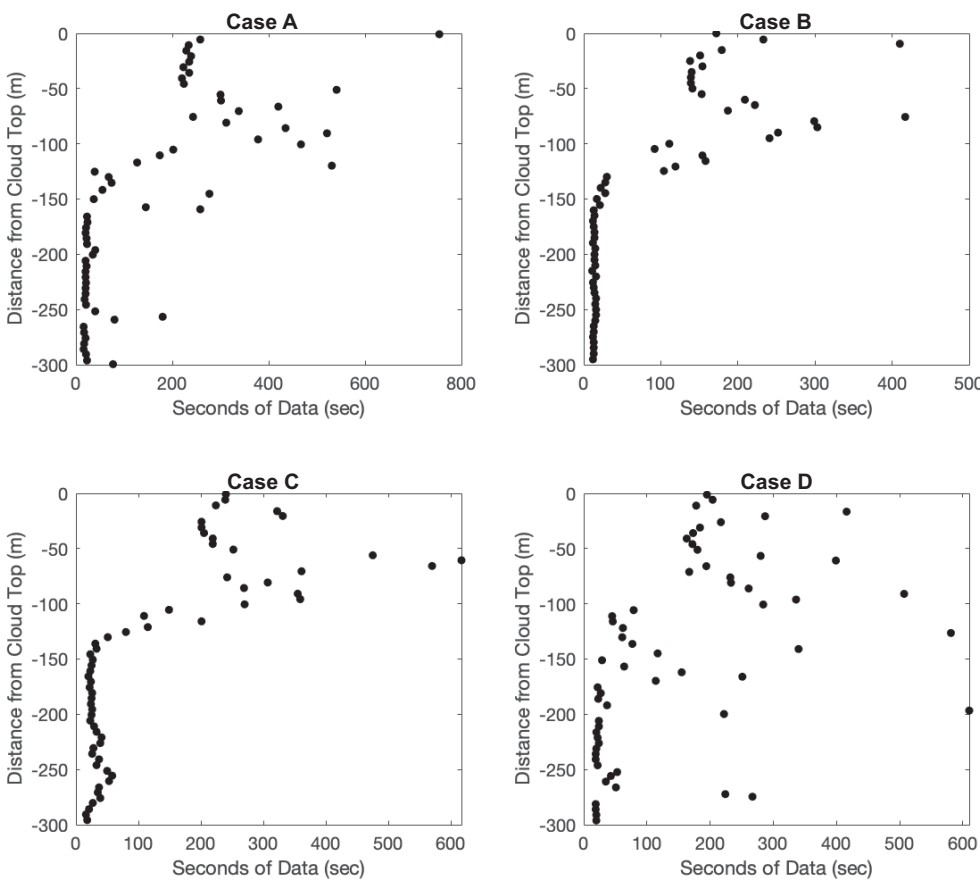

**Figure 3.** Amount of data for four of the five POST flights selected for additional analysis. Points represent seconds of non-zero data within each 5 m $z_{\text{shift}}$ altitude bin and are concentrated within $\sim$150 m of cloud top.

### 3.3.1 Case A: POST day 2008/07/30

Case A (Fig. 4) is an example of the best case scenario for agreement between MODIS and in situ profiles. The PDI observed number concentration (open circles) calculated at mid-cloud is $N_{\text{d}} = 40 \pm 2\,\text{cm}^{-3}$, while MODIS estimates $N_{\text{d}} = 38 \pm 4\,\text{cm}^{-3}$

(solid red line). The satellite assumes that this number concentration is fixed throughout the vertical cloud profile. In reality $N_{\text{d}}$ is not constant with height, but instead gradually drops off towards cloud base and cloud top. We attribute lower values of $N_{\text{d}}$ near cloud top to cloud drop evaporation due to turbulent entrainment of warm, dry air from above the boundary layer. Low values near cloud base may be an effect of uneven cloud base, which affects the altitude range over which activation causes $N_{\text{d}}$ to increase with height. In addition, cloud base is typically higher in those portions of the cloud layer that experience

downdrafts due to lower adiabaticity (Zhou and Bretherton, 2019), which can affect $N_{\text{d}}$ in the cloud base region. Through

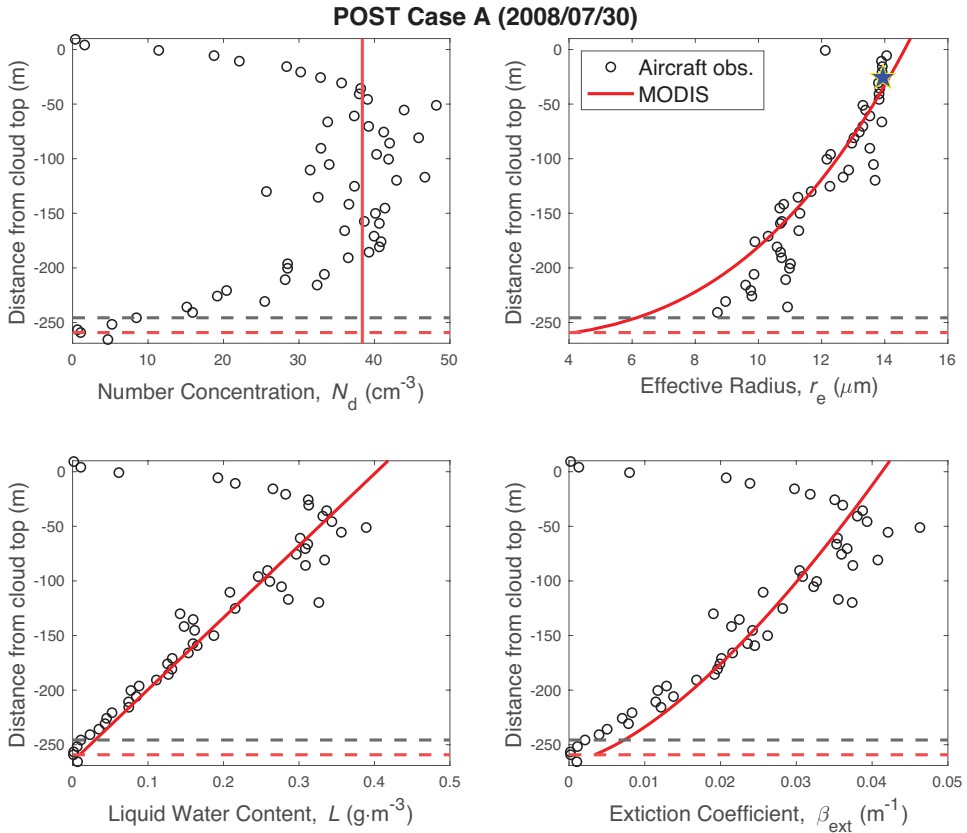

**Figure 4.** Profiles of $N_d$, $r_e$, $L$, and $\beta$ for POST Case A (2008/07/30) comparing aircraft data (circles) and MODIS retrieval profiles (red lines). The aircraft measured cloud base is the black dashed line while the MODIS estimated cloud base is the red dashed line. Aircraft-derived $r_e$ that most closely corresponds to the MODIS cloud top value is indicated by the blue star, and is placed at an altitude determined by the max weight from Eq. 3. MODIS retrieval profiles of $N_d$, $r_e$, $L$, and $\beta$ are based on Eqs. 1, 10, 7, and 11, respectively.

the middle of the cloud $N_d$ does appear reasonably constant on average. Because constant $N_d$ is consistent with the MODIS assumption, we use this as justification for the use of mid-cloud level legs for the MODIS-PDI $N_d$ comparison.

The vertical profiles of $L$, $r_e$, and $\beta$ all agree very closely during this flight. The MODIS estimate of cloud top $r_e$ (blue star) is plotted at an altitude determined by the weighting function (Eq. 3) and is consistent with aircraft measurements. Lastly, cloud base altitude as estimated from MODIS and aircraft (red and black dashed lines, respectively) also agree well. The alignment between this full set of MODIS and aircraft observations results in an accurate estimate of $N_d$ as long as we interpret this value as a mid-cloud estimate. However, this day represents only two of the flights where agreement across all quantities occurs.

### 3.3.2 Case B: POST day 2008/07/21

Unlike the strong agreement between MODIS and PDI seen in Case A, Case B (Fig. 5) illustrates a day in which all of the profiles have very poor agreement. The MODIS retrieved number concentration is $N_d = 42 \pm 12$ cm$^{-3}$ while the in situ

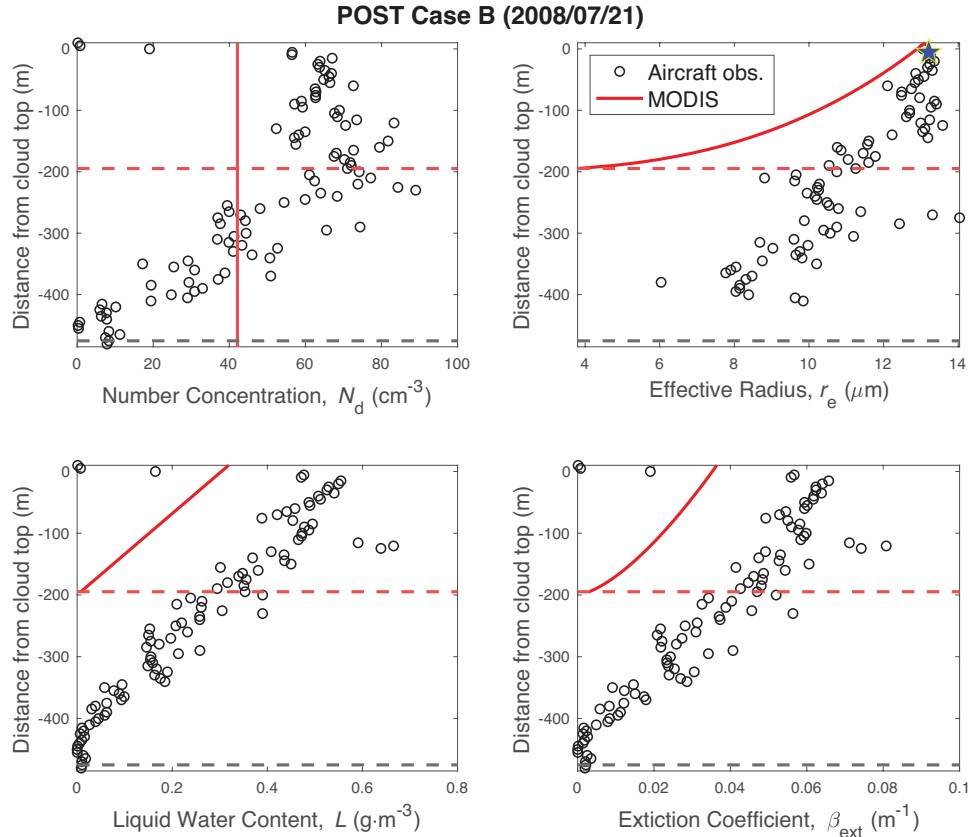

**Figure 5.** Comparison of profiles of $N_d$, $r_e$, $L$, and $\beta$ from aircraft observations (circles) and MODIS retrievals (red lines) for POST Case B (2008/07/21). See Fig. 4 for more details.

value is $N_d = 63 \pm 3$ cm$^{-3}$. This disagreement occurs despite the fact that the satellite cloud top effective radius (blue star) is well-matched to the PDI estimate ($13.0 \pm 0.9$ μm vs. $13.0 \pm 0.5$ μm), indicating that cloud top $r_e$ is not the source of the $N_d$ disagreement.

While the MODIS and PDI $r_e$ profiles agree at cloud top, at all other altitudes MODIS greatly underestimates effective 295 radius. All else being equal, this should lead to an overestimation of $N_d$, which is not what we see. Instead, we attribute the MODIS $N_d$ underestimation to disagreement with the MODIS $\tau_c$ retrieval. Any error in MODIS $\tau_c$ propagates to both the $\beta$ and $L$ profiles. Integration of the MODIS $\beta$ profile results in a $\tau_c$ value smaller than observed, leading to an underestimation of $N_d$ ($N_d \propto \tau_c^{1/2}$) as seen in Fig. 5, as well as a cloud base that is too high, which is equivalent to a cloud that is too thin.

### 3.3.3 Case C: POST day 2008/07/17

During POST flight 2008/17/07, the MODIS number concentration is $N_d = 260 \pm 31$ cm$^{-3}$, almost twice the PDI value of $N_d = 140 \pm 5$ cm$^{-3}$ (Fig. 6). In contrast to Case B, we attribute the cause of this disagreement to MODIS retrieved cloud top

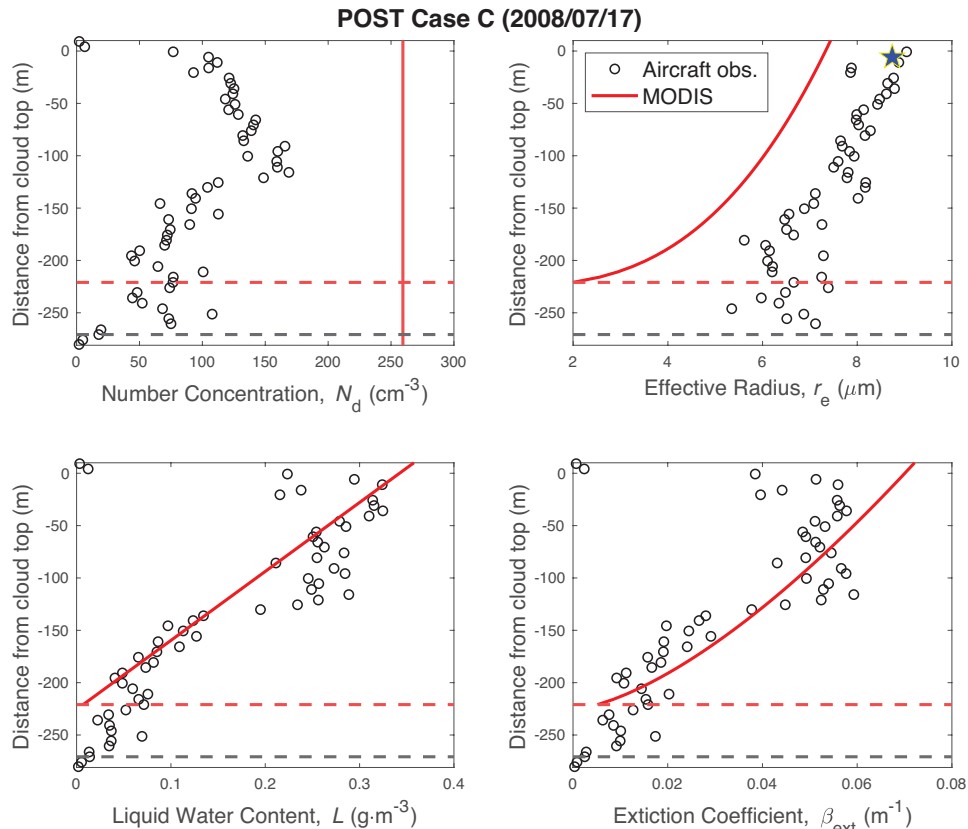

**Figure 6.** Comparison of profiles of $N_d$, $r_e$, $L$, and $\beta$ from aircraft observations (circles) and MODIS retrievals (red lines) for POST Case C (2008/07/17). See Fig. 4 for more details.

$r_e$. The $\beta$ and $L$ profiles are in close agreement, presumably because MODIS and PDI-derived $\tau_c$ are in agreement. However, both the PDI cloud top $r_e$ as well as the PDI $r_e$ vertical profile are greater than the MODIS estimates, leading to the large overestimation of $N_d$.

### 305   3.3.4   Case D: POST day 2008/08/04

MODIS overestimates number concentration during POST Case D (2008/08/04), finding an $N_d = 180 \pm 12$ cm$^{-3}$ compared to a PDI value of $N_d = 122 \pm 5$ cm$^{-3}$ (Fig. 7). This case illustrates a day when the assumption that $N_d$ is constant with altitude is not accurate, with the retrieval overestimating $N_d$ at all altitudes. The $r_e$ subplot shows that there is good agreement between PDI and MODIS in both the cloud top $r_e$ value as well as the $r_e$ vertical profile (in the region of most influence, i.e. within 310   50 m of cloud top). However, the MODIS $\beta$ and $L$ profiles differ to a substantial degree from PDI. The MODIS $\beta$ profile is greater than what is observed, meaning that the MODIS $\tau_c$ is also greater than the PDI value. Due to the relationship between

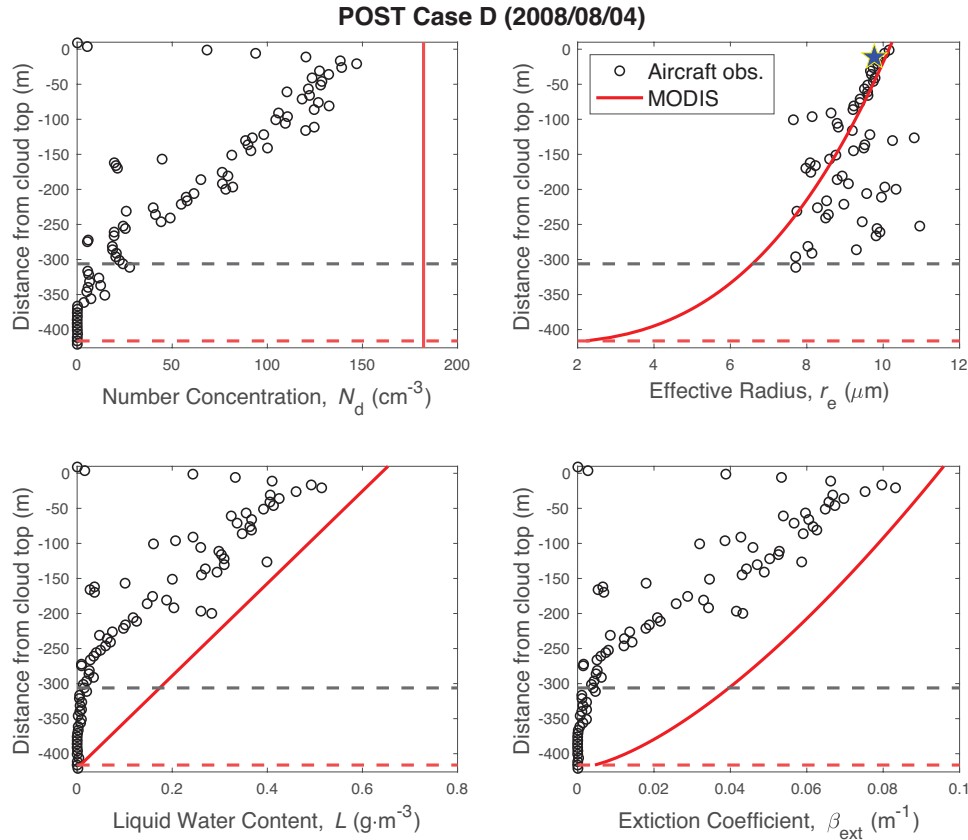

**Figure 7.** Comparison of profiles of $N_d$, $r_e$, $L$, and $\beta$ from aircraft observations (circles) and MODIS retrievals (red lines) for POST Case D (2008/08/04). See Fig. 4 for more details.

$\tau_c$ and $N_d$, it follows that an overestimation in MODIS optical depth should lead to an overestimation in $N_d$ which is indeed what we see in Fig. 7.

### 3.3.5    Case E: POST day 2008/08/14

POST Case E (2008/08/14) illustrates the degree of sensitivity in the MODIS $N_d$ retrieval. Although the agreement between the profiles of $r_e$, $L$, and $\beta$ is not perfect, it is reasonably close (Fig. 8). Despite this, MODIS significantly overestimates number concentration ($N_d = 98 \pm 5$ cm$^{-3}$ compared to the PDI value of $N_d = 71 \pm 4$ cm$^{-3}$). Even small errors in effective radius manifest as a large error in number concentration, presumably due to the non-linear relationship between the two. This suggests that satellite profile estimates must be fairly accurate in order to successfully retrieve $N_d$.

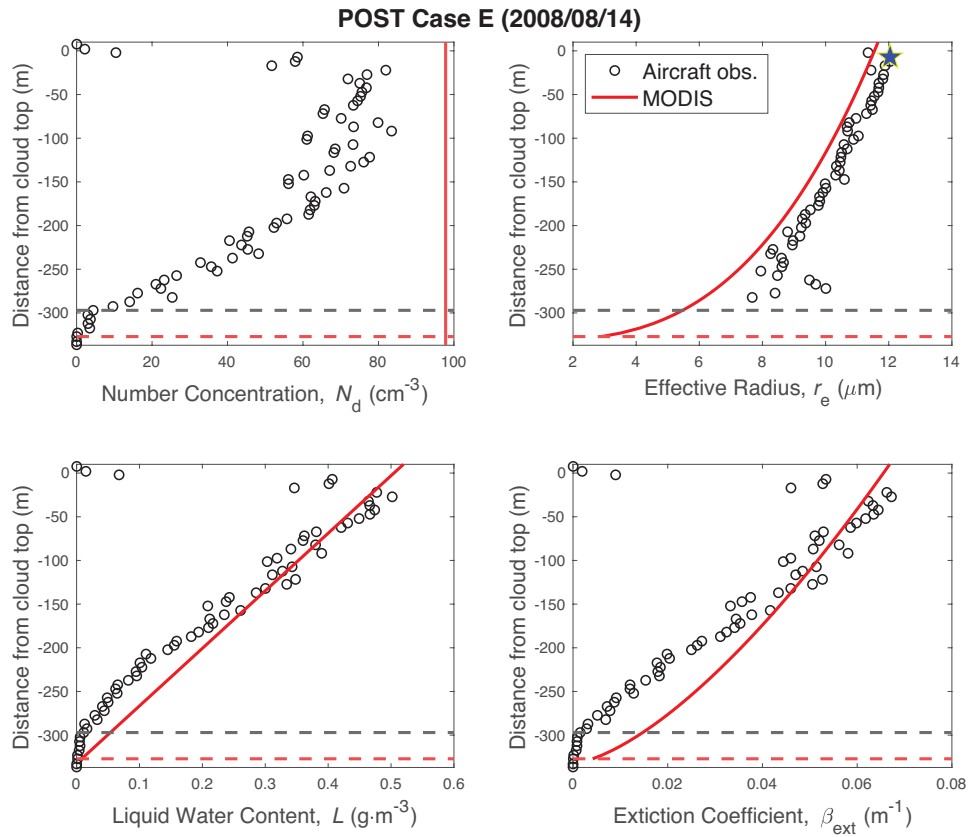

**Figure 8.** Comparison of profiles of $N_d$, $r_e$, $L$, and $\beta$ from aircraft observations (circles) and MODIS retrievals (red lines) for POST Case E (2008/08/14). See Fig. 4 for more details.

### 3.3.6 Summary of POST Cases

Based on POST profile analysis of the important variables that determine MODIS number concentration, we find that there are cases in which all variables agree well with PDI observations as well as cases where one or more variables disagree. However, out of all eight days considered, there are no cases in which the MODIS $N_d$ was correct due to compensating errors in the underlying variables (i.e. $\beta$, $L$, $r_e$). If the satellite number concentration is a match to the in situ value, it is due to correct estimations in all variables. Conversely, if even one variable disagrees with observation, $N_d$ is also inaccurate. Fairly good agreement in all three profiles can also still yield significant discrepancy in $N_d$.

### 3.4 Analysis of $k$ and $f_{ad}$

The accuracy of the MODIS assumptions concerning $f_{ad}$ and $k$ can also affect retrieved number concentration. The MODIS retrieval assumes that $f_{ad}$ has a constant value of 0.6. If $f_{ad}$ were significantly different from this assumed value, we would find

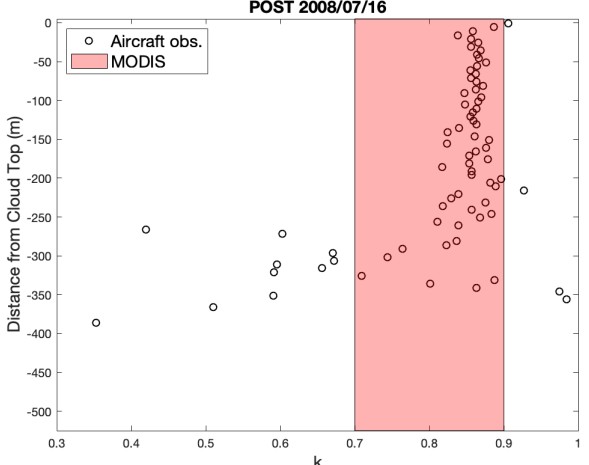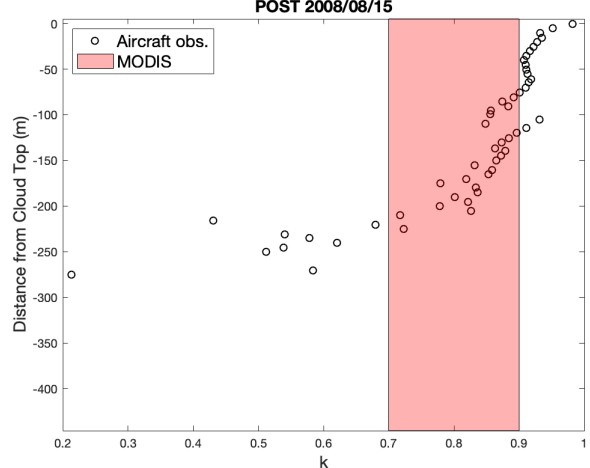

**Figure 9.** Comparison of $k$ values from the POST campaign for days 2008/07/16 and 2008/08/15 with the MODIS assumed range of $0.7 \leq k \leq 0.9$ (red shading). The PDI values are calculated using Eq. 6 and represent flight averages at each 5 m $z_{\text{shift}}$ bin.

that the assumed MODIS slope of $L(z)$ would disagree with observations. We find that for each of the 8 POST flights, the in situ $L(z)$ slopes are well matched to MODIS, even if the absolute values are in disagreement. This leads us to conclude that $f_{\text{ad}}$ has little effect on the $N_{\text{d}}$ retrieval for the cases that we analyzed.

The MODIS $N_{\text{d}}$ retrieval assumes $k$ is a constant of value $0.8$. Analysis by Grosvenor et al. (2018) concludes that $k$ ranges from 0.7 to 0.9 (see also Lebsock and Witte, 2023). We evaluate this conclusion using the POST data (two examples shown in Fig. 9). Through the bulk of cloud profiles, the MODIS algorithm assumption is found to be generally reasonable. An uncertainty in $k$ of $\pm 0.1$ will lead to an uncertainty in $N_{\text{d}}$ of 10 to 15%. Near cloud base, $k$ can be much smaller. However, this can be considered inconsequential, as this region contributes little to the MODIS retrievals. There are also cases in which $k$ exceeds this range near cloud top. However, occasional outlying behavior should not have a large impact on the retrieval and by default $N_{\text{d}}$ as well.

## 3.5 Determination of Cloud Base

As illustrated by Fig. 5, the large inaccuracies in the MODIS profile assumptions for POST Case B are accompanied by a large satellite overestimation of cloud base altitude (over 200 m difference from observed). Fig. 10 compares MODIS derived cloud base to the observed cloud base altitude for all eight of the POST flights. MODIS estimates cloud base height within 50 m of the aircraft observations in 5 of the 8 cases. For the remaining three cases, MODIS cloud base differs by over 100 m, which is particularly notable since this is a significant fraction of the cloud depth. Due to the small sample size, however, it is difficult to quantify the frequency of accurate MODIS estimations of cloud base.

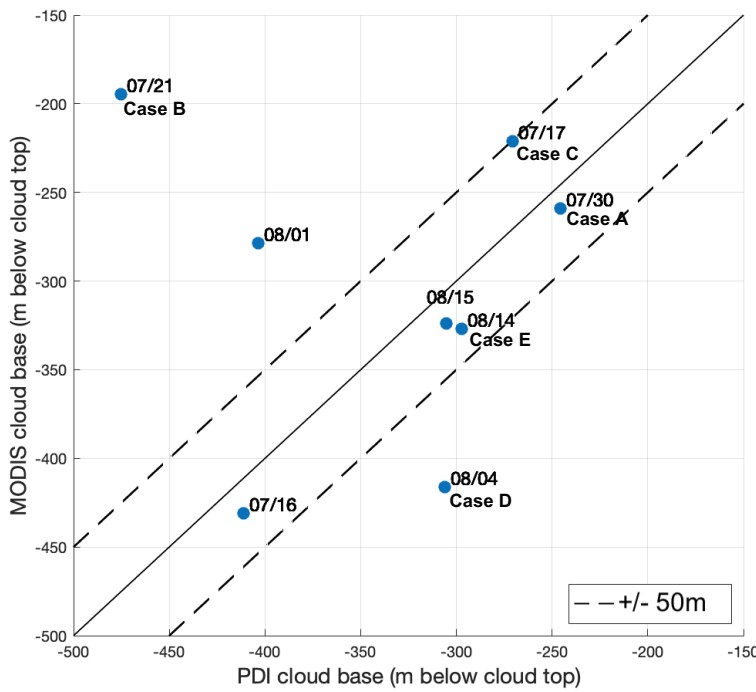

**Figure 10.** Comparison of MODIS and aircraft-estimated cloud base altitude for each day of the POST campaign. The 1:1 line (solid), and $\pm 50$ m lines (dashed) are also shown.

## 4 Conclusions

In this study, we compare $N_{\mathrm{d}}$ derived from MODIS products with in situ observations recorded by the PDI instrument over three different field campaigns (MASE, VOCALS, and POST) sampling marine stratocumulus clouds. We also compare cloud

microphysical and radiative variables relevant to the calculation of $N_{\mathrm{d}}$ using observations from the POST campaign. These variables include vertical profiles of $r_{\mathrm{e}}$, $L$, and $\beta$.

Our results show that while there are instances in which the MODIS retrieval predicts number concentration within sampling variability, there are a significant number of cases in which the satellite overestimates $N_{\mathrm{d}}$. Our results suggest that the discrepancy between retrieved and in situ $N_{\mathrm{d}}$ can be greater than $\pm 50\%$, roughly in line with the $\pm 80\%$ overall uncertainty previously

proposed (Bennartz, 2007; Grosvenor et al., 2018). We find that the apparent overestimation bias in number concentration does not originate as a bias in MODIS $r_{\mathrm{e}}$. This is consistent with the conclusions of Witte et al. (2018) who found no obvious bias between MODIS and PDI derived effective radius. We also find that it is possible for $N_{\mathrm{d}}$ to be accurately retrieved with a poor retrieval of $r_{\mathrm{e}}$, presumably due to compensating errors in other retrieval parameters.

Two out of the 8 POST cases studied exhibit good agreement in $N_d$, as well as in the profiles of $r_e$, $L$, and $\beta$. For the remaining six cases, we do not attribute $N_d$ discrepancy to a single error source. Instead, we show that there are several different cases which result in incorrect number concentration:

1. MODIS incorrectly predicts profiles of $r_e$, $L$, and $\beta$

2. MODIS incorrectly predicts profiles of $L$ and $\beta$ but accurately estimates $r_e$ (either the full $r_e$ profile or at the most influential altitude near cloud top)

3. MODIS incorrectly predicts profiles of $r_e$ but accurately estimates $L$ and $\beta$ profiles

We also show a case where all profiles appear to exhibit reasonably good but imperfect agreement, but the resulting $N_d$ does not agree well at all due to compounding errors.

Accurate $N_d$ retrievals are representative of mid-cloud values. Although MODIS assumes number concentration is constant through a cloud's vertical profile, even in the best case scenarios it appears to be a poor reflection of cloud top and cloud base conditions. We also show one case where the assumption that $N_d$ is constant with altitude is not accurate.

In order to improve the MODIS $N_d$ retrieval, it would be beneficial to acquire more data collected in a manner similar to the POST campaign, i.e., with repeated sawtooth-like penetrations through cloud top. Multiple profiles near cloud top allow for deeper analysis of the underlying variables in the retrieval which can be used to more accurately quantify sources of error.

*Data availability.* PDI data from POST and VOCALS are freely available at https://data.eol. ucar.edu, and from MASE at https://doi.org/10.5281/zenodo.1035928. MODIS Level 2 retrievals are available from https://search.earthdata.nasa.gov.

*Author contributions.* SRP performed the data analysis and wrote the manuscript. MKW and PYC designed the study concept, obtained data, and edited the manuscript.

*Competing interests.* The authors declare no competing interests.

*Acknowledgements.* We thank the CIRPAS Twin Otter pilots, crew and instrument scientists for their efforts to collect the data used in this study. MKW acknowledges support from the Office of Naval Research Marine Meteorology program under funding documents N0001424WX00817 and N0001424WX02198.

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
