# Peer review of "Aircraft Evaluation of MODIS Cloud Drop Number Concentration Retrievals"

_Atmospheric Measurement Techniques, 2024_

## Referee Comment (RC1)

**Review of *Aircraft Evaluation of MODIS Cloud Drop Number Concentration Retrievals* – Passer et al. (2024)**

This paper evaluates MODIS-retrieved concentrations of cloud drops to those measured in situ based on three field campaigns that leveraged the CIRPAS Twin Otter aircraft. The results reveal that in some cases, the MODIS retrievals are within the range of uncertainty in the aircraft measurements, which coincides with all parameters used in the MODIS retrieval also agreeing well with the in situ measurements. However, in other cases, MODIS deviates considerably from the observed concentrations, and the authors do not find any cases where the MODIS-retrieved droplet number concentration is correct due to compensating errors in the retrieval parameters. The authors present different reasons for these errors based on the variables used in the retrieval algorithm.

In general, this paper is well written and easy to follow. I believe it will make a solid addition to the literature and provides a foundation for understanding uncertainty and errors in MODIS-retrieved droplet concentrations, which are commonly used in the field.

Other than some minor comments/suggestions to help clarify some sentences, my only major comment is whether the authors have considered an analysis of the different cases in terms of other variables to determine the conditions under which MODIS retrievals are more or less accurate? More specifically, I am thinking about variables not used in the retrieval but could indicate conditions in which the MODIS retrievals would be more/less accurate.

If the authors have any questions, please do not hesitate to reach out!

Zachary J. Lebo

Major Comment

1) As denoted above my only somewhat major comment is regarding an expansion of the analysis to include other variables that may help determine why MODIS is biased in its droplet number concentration retrieval in some instances and not others. I think a first go at this could just be re-doing Figs. 1 and 2 to be color coded not by day but by another variable or the difference between the MODIS and in situ values. The latter might just be a cleaner way to demonstrate some of the later analysis. Beyond the variables used in the retrieval, are there other in situ observations that you could use to "color" the points in these figures and help discern conditions that MODIS over- or underestimates the droplet number concentration? Thermodynamics? Other characteristics of the drop size distribution (skewness, bimodality)? Pitot tube measurements of eddy dissipation? Just some random thoughts. These may all very well show no relation to the MODIS biases, but I think it is worth the effort to show this.

Minor Comments

1) Lines 80-81: How does this compare to the in situ measurements? At least based on Fig. 1 using the standard error calculations, in many cases, the error bars for the PDI are larger than those of the MODIS retrievals. Granted these are different things, but just want to be sure that it is recognized in the paper that MODIS and the in situ observations come with their own uncertainty.
2) Lines 91-94: Why just these three campaigns? Is it a limit of the PDI being used?
3) Line 97: Suggest adding 1-3 sentences briefly describing this matching.
4) Lines 107-108: data "are".
5) Lines 108-109: This is rather vague. What do you mean by "most representative"? In terms of what?
6) Line 111: I am not understanding the "are selected during flight by the flight scientist" part. Can this be omitted?
7) Line 114: Should this be a relative distance or does it not matter for these thin clouds? For deeper clouds, 60-90 m from cloud top, in my opinion, would still be cloud top.
8) Lines 154-155: How do you quantify "very good agreement"?
9) Fig. 1: In the figure, N is used for the concentration but $N_d$ is used in the text.
10) Line 214: No idea why I put this comment on this line, but one thing that I noticed is that from Case 1, to Case 2, and then Case 3, the droplet concentration increased. Is that consistent? Statistically, do the cases with the lowest concentrations agree best with MODIS? Why?
11) Line 214: "the" MODIS number concentration.
12) Line 214: "…almost twice the PDI…"
13) Lines 256-257: Do you mean the MODIS-retrieved cloud base is within 50 m of the in situ observations? I interpreted this as the MODIS retrieval footprint location at first.
14) Lines 258-259: Is there a better (other) way to test MODIS-retrieved cloud base than using aircraft?

---

## Author Comment (AC1)

**Response to Reviewers**

**REVIEWER 1**

Review of *Aircraft Evaluation of MODIS Cloud Drop Number Concentration Retrievals* – Passer et al. (2024)

This paper evaluates MODIS-retrieved concentrations of cloud drops to those measured in situ based on three field campaigns that leveraged the CIRPAS Twin Otter aircraft. The results reveal that in some cases, the MODIS retrievals are within the range of uncertainty in the aircraft measurements, which coincides with all parameters used in the MODIS retrieval also agreeing well with the in situ measurements. However, in other cases, MODIS deviates considerably from the observed concentrations, and the authors do not find any cases where the MODIS-retrieved droplet number concentration is correct due to compensating errors in the retrieval parameters. The authors present different reasons for these errors based on the variables used in the retrieval algorithm.

In general, this paper is well written and easy to follow. I believe it will make a solid addition to the literature and provides a foundation for understanding uncertainty and errors in MODIS retrieved droplet concentrations, which are commonly used in the field.

Other than some minor comments/suggestions to help clarify some sentences, my only major comment is whether the authors have considered an analysis of the different cases in terms of other variables to determine the conditions under which MODIS retrievals are more or less accurate? More specifically, I am thinking about variables not used in the retrieval but could indicate conditions in which the MODIS retrievals would be more/less accurate.

If the authors have any questions, please do not hesitate to reach out!

Zachary J. Lebo

Major Comment

> 1) *As denoted above my only somewhat major comment is regarding an expansion of the analysis to include other variables that may help determine why MODIS is biased in its*

*droplet number concentration retrieval in some instances and not others. I think a first go at this could just be re-doing Figs. 1 and 2 to be color coded not by day but by another variable or the difference between the MODIS and in situ values. The latter might just be a cleaner way to demonstrate some of the later analysis. Beyond the variables used in the retrieval, are there other in situ observations that you could use to "color" the points in these figures and help discern conditions that MODIS over- or underestimates the droplet number concentration? Thermodynamics? Other characteristics of the drop size distribution (skewness, bimodality)? Pitot tube measurements of eddy dissipation? Just some random thoughts. These may all very well show no relation to the MODIS biases, but I think it is worth the effort to show this.*

We agree with the reviewer and like this idea but think this further exploration will be reserved for a follow-up study.

Minor Comments

1) *Lines 80-81: How does this compare to the in situ measurements? At least based on Fig. 1 using the standard error calculations, in many cases, the error bars for the PDI are larger than those of the MODIS retrievals. Granted these are different things, but just want to be sure that it is recognized in the paper that MODIS and the in situ observations come with their own uncertainty.*

   This line was corrected, as we mis-interpreted the uncertainty from Grosvenor et al. To be clear, their estimated uncertainty is 78%, consistent with Bennartz (2007) theoretical analysis suggesting 80% uncertainty.

   We have added a paragraph at the end of Section 2.1.2 to discuss in more detail the uncertainties associated with the in situ observations.

2) *Lines 91-94: Why just these three campaigns? Is it a limit of the PDI being used?*

   That's correct. We limited our analysis to campaigns with well-characterized PDI observations because we are more confident in the estimate or $r_e$. Although PDI data is also publicly available from ORACLES, we are aware of some significant issues in the post-processing performed by the instrument PI that prevent accurate calculation of cloud optical depth. Another recently collected PDI dataset from two of the co-authors (MKW and PYC) was not available when the first author performed this work.

3) *Line 97: Suggest adding 1-3 sentences briefly describing this matching.*
   The relevant sections (2.1 and 2.2) were rewritten to give a brief description of the matching procedure as well as details on the MODIS products used in the study.

4) *Lines 107-108: data "are".*
   Corrected

5) *Lines 108-109: This is rather vague. What do you mean by "most representative"? In terms of what?*
"Most representative" of what MODIS would retrieve, which is something like a vertical mean value, since $N_d$ near cloud top is often impacted by entrainment, and thus not representative of the entire cloud. Edited to "is more representative of the mean cloud $N_d$ value relative to cloud top values (as will be shown below)."

6) *Line 111: I am not understanding the "are selected during flight by the flight scientist" part. Can this be omitted?*
Great suggestion, done

7) *Line 114: Should this be a relative distance or does it not matter for these thin clouds? For deeper clouds, 60-90 m from cloud top, in my opinion, would still be cloud top.*
This line was updated. We did not mean to imply that 60-90 m below cloud top should be considered "mid-cloud." Rather, when the aircraft was primarily using sawtooth sampling vs. level legs, the sawtooth pattern typically only penetrated about 100 m below cloud top so we use the 60-90 m range as an analog to level mid-cloud legs to characterize $N_d$.

8) *Lines 154-155: How do you quantify "very good agreement"?*
Great suggestion, done

9) *Fig. 1: In the figure, N is used for the concentration but $N_d$ is used in the text.*
Figure 1 was updated accordingly.

10) *Line 214: No idea why I put this comment on this line, but one thing that I noticed is that from Case 1, to Case 2, and then Case 3, the droplet concentration increased. Is that consistent? Statistically, do the cases with the lowest concentrations agree best with MODIS? Why?*

Based on the small sample used here, we don't see evidence for a statistically robust relationship between Nd and satellite-in situ agreement (see Fig. 1). Pure coincidence that the order in which the POST cases were presented hinted at such a progression.

11) *Line 214: "the" MODIS number concentration.*
Added

12) *Line 214: "...almost twice the PDI..."*
Corrected

13) *Lines 256-257: Do you mean the MODIS-retrieved cloud base is within 50 m of the in situ observations? I interpreted this as the MODIS retrieval footprint location at first.*
That's correct (cloud base). Sentence revised to:
"MODIS estimates cloud base height within 50 m of the aircraft observations in 5 of the 8 cases."

14) Lines 258-259: Is there a better (other) way to test MODIS-retrieved cloud base than using aircraft?

Definitely! Active remote sensors are great for characterizing cloud vertical boundaries. Unfortunately, they can't provide the detailed microphysical characterization we obtain from in situ probes nor was the aircraft equipped with any of these during the flights we examine. To be internally consistent, we work with what we've got – in situ probe estimates of cloud base from aircraft vertical profile maneuvers.

*This study evaluates MODIS cloud droplet number concentration (Nd) retrievals by comparing them to in situ measurements from three aircraft campaigns. The results indicate that MODIS tends to overestimate Nd, with discrepancies of 50% or more. The authors explore potential sources of retrieval errors and conclude that no single factor dominates the discrepancies. This study is a valuable contribution to the validation of satellite-derived cloud microphysical properties, although the limited number of cases constrains the statistical conclusions.*

*I recommend publishing this manuscript, as it presents valuable findings. The study is well-structured, and the results contribute to our understanding of limits and merits of satellite-based cloud's microphysical retrievals. However, the authors may consider the following comments and suggestions to further strengthen their analysis and discussion:*

*The PDI is treated as the reference dataset in this study, but systematic biases in PDI measurements should be acknowledged and discussed. The authors may consider exploring how potential biases in the PDI might influence the comparison with MODIS retrievals, particularly in the context of cloud droplet number concentration estimates. A discussion of known uncertainties in PDI data and their implications for the conclusions drawn in this study would enhance the robustness of the analysis.*

We have added a section on PDI uncertainties (see Section 2.1.2, Lines 189-200).

*The introduction would benefit from a more detailed discussion of past validation studies on cloud droplet number concentration retrievals. A clearer connection to previous validation efforts would help contextualize this work within the broader field of cloud microphysics and remote sensing. The authors may also discuss how their findings compare to and extend prior research, highlighting the specific advancements made in this study.*

The introduction has been greatly revised and expanded.

*The authors may investigate whether there is a correlation between the bias in cloud base and other key parameters, such as the optical depth or the profile of liquid water content. If such correlations exist, they could provide insights into the nature of systematic retrieval biases and their physical causes. Addressing this point could strengthen the interpretation of the results and provide a more comprehensive view of potential error sources.*

There is inevitably a correlation among the retrieved cloud base and the vertical profiles due to the fact that the MODIS retrieval only has two parameters, $r_{eff}$ and $\tau_c$. In simple terms, $r_{eff}$ pins cloud top properties, and $\tau_c$ is used to extrapolate downwards to cloud base. So all the profiles of LWC, $\beta$, and $r_{eff}$, plus cloud base, have to be correlated to some degree due to the lack of degrees of freedom for them to be independent. And thus bias in one of these will be correlated to a bias in

another. It's unclear whether further analysis will provide any new information, so we elected not to dig more deeply into this question.

*Moreover, entrainment can affect the assumptions made about both k and f. The authors may consider expanding their discussion on how entrainment variability influences these quantities and whether it introduces systematic biases in retrievals. A quantitative or qualitative assessment of entrainment's impact on the assumed values would provide additional clarity. Given that entrainment may contribute to the observed mismatch in retrieved quantities, it might be beneficial to attempt an estimate of entrainment at the cloud top using MODIS data. Specifically, high spatial variability in brightness temperature or effective radius over some area of the cloud deck could serve as a proxy for entrainment activity.*

*The authors may consider evaluating whether this estimated entrainment degree correlates with biases in the retrieved quantities, which could provide further evidence of the role of mixing in retrieval uncertainties.*

We do find this question interesting. However, the way our analysis is performed, we actively try to avoid the entrainment region as much as possible (with the exception of $r_{eff}$, which is always in the cloud top region where entrainment has its greatest influence). Or put another way, our conclusion is that MODIS retrieved $N_d$ is most suitable to describe the middle of the cloud where entrainment is less influential.

That said, there could very well be a bigger picture research question of how entrainment influences not just cloud top, but the entire cloud layer, and thus may play some role (small, medium or large) in determining when MODIS retrievals are accurate or not. However, we deemed this to be outside the scope of this manuscript, and instead a consideration for subsequent studies!

*These comments are intended to refine the discussion and improve the clarity of the manuscript. Overall, I find this study to be a valuable contribution to the field and recommend its publication. I believe the manuscript remains suitable for publication even if not all the above suggestions are fully met.*

**REVIEWER 3**

The retrieval of cloud microphysical properties from passive visible-to-infrared satellite measurements is characterized by large uncertainties, and comparisons with in situ observations typically show limited agreement. In this study the authors make an attempt to attribute errors in satellite-retrieved cloud droplet number concentration ($N_d$) to errors in several underlying retrieval variables. The results show that there is a variety of reasons why the $N_d$ retrievals can be off. The analysis is a useful contribution to the scientific literature.

General comments

*Partly due to the limited number of cases the results are for the most part statistically inconclusive. I wonder if there are more aircraft measurements available that could be included, e.g. from the ORACLES campaign. In addition, it would be good to include a discussion on possible biases in the PDI measurements, since these are used as reference. Meyer et al. (2024) recently found systematic differences between PDI and other in situ probes, which could be referred to.*

We agree with the reviewer that the number of cases available for analysis is quite limiting. This is an unfortunate fact of dealing with aircraft data. Our use of data from more than one campaign already sets us apart from most prior studies; we are aware that our exclusive use of PDI data prevents us from analyzing as broad a range of cases as, e.g., Gryspeerdt et al. (2022).

Re: ORACLES
The measurements used in this study from MASE, POST and VOCALS were re-processed after it was recently discovered that PDI sample volume (a quantity needed to convert droplet counts to concentration) varies with detected drop size, resulting in some older reported measurements being inaccurate. Mayer et al. (2024) were able to use effective radius in their study because sample volume cancels out, but for computing optical depth, number concentration and liquid water content, that is not the case. One of the co-authors of this study (MKW) was also a co-author of the Mayer et al. (2024) paper and is working with the ORACLES measurement PI to obtain raw PDI data for reprocessing. Unfortunately, this work has not yet been completed.

Re: broader discussion of differences among probes
First, we stress that differences among probes should not be considered "biases" as there is no in situ "truth" value to which measurements can be compared. Legacy probes are not more accurate simply because they've been around longer. Witte et al. (2018) found systematic differences in $r_e$ between PDI and forward-scattering-based cloud probes with the same basic takeaway as Mayer et al. (2024): PDI measures values of cloud drop effective radius a few μm larger than forward-scatterers. Witte et al. (2018) suggest that this occurs because PDI detects droplets in the approximate size range 30<d<80 μm more efficiently than either forward-scattering probes (FSSP, CDP, CAS) or optical array probes (CIP, 2DS, etc.), but this assertion has not been rigorously tested. Lebsock & Witte (2023, see their Fig. 2) compare histograms of number concentration, liquid water content, effective radius, and the k parameter (effectively a measure of drop size distribution width) across different instruments, but these measurements are from different field experiments and therefore can't be directly compared.

In our opinion, a systematic evaluation of airborne microphysical instruments is long overdue given the recent maturation of several technologies (of course, PDI; also, holographic detectors and the new generation of optical array probes such as the 2DS that detect droplets d<100 μm). But given that the data for such an intercomparison does not yet exist, it is beyond the scope of this publication to broach the topic.

Specific comments

*P1, L15-16: This sentence reads strange: 'cloud properties ..., such as cloud radiative effects, precipitation, and aerosol-cloud interactions.'. These are not really cloud properties but effects of clouds.*

The introduction was rewritten and this phrasing is no longer used.

*Section 1.1 is better placed in the Methods section rather than in the Introduction. The Introduction should also be extended by embedding the study in the existing literature, e.g., referring to earlier validation studies, and adding appropriate references.*

We chose to keep section 1.1 in the Introduction as we felt the logical flow was more natural when introducing the retrieval framework separate from the observations and analysis methods. We did, however, incorporate the reviewer's recommendations to contextualize our study with respect to existing literature in the rewritten introduction.

*Section 1.1: No details are given about the origin of the MODIS tau_c and r_e retrievals. Are these the 'standard' MODIS products (MOD06/MYD06 C6.1)? Which satellites (Terra or Aqua) were used for which cases, and did the authors analyse whether there was a (systematic) difference between them? Which shortwave-infrared channel was used? Was it 3.7 micron? These details are very important and should be included in the manuscript.*

Details are now given in Section 2.2 ("Satellite retrieval details and sampling methodology"):
"We utilize MODIS collection 6.1 retrievals of $r_e$ and $\tau_c$ using the 2.1 μm band ("Cloud_Effective_Radius" and "Cloud_Optical_Thickness" products, respectively), consistent with Witte et al. (2018). Over the small sample size considered here, we find no evidence of systematic differences between satellites."

*The writing in Section 1.1 is slightly sloppy and must be improved. Here are some examples:*

*- The meaning of the symbols in Eq. (1) should be presented after the equation and not much later. Of course, further explanation can follow later.*

All variables are now explained immediately after Eq. (1) is introduced.

*- It would be good to include the definition of effective radius, r_e, in this section.*

Added

*- 'In order to estimate r_e, MODIS makes use of a weighting function': this is not how it really works. The MODIS retrievals use shortwave infrared radiance measurements to infer r_e. And since the radiance in these channels originates mainly from near the cloud top (as quantified by the weighting function), the retrieval is representative of r_e in that region near the cloud top.*

Indeed. The wording here was incorrect before and has been updated to accurately reflect our motivation for employing the weighting function:
"To produce an estimate of $r_e$ from aircraft suitable for comparison with that from MODIS, a weighting function is used to weight the impact of cloud vertical structure on the aircraft-derived variables"

*- In Eq. (2), optical depth tau without a subscript appears. What is it?*

All instances of tau now have a subscript c.

*- 'mu and mu0 depend on satellite position and correspond to the solar zenith angle and sensor zenith angle, respectively.'. Normally mu0 denotes the solar zenith angle. Also the solar zenith*

*angle does not depend on the satellite position.*
Updated the phrase in question to read:
"mu and mu0 are the cosine of the sensor and solar zenith angles, respectively"

*- Eq. (4): r is not defined. Q_ext depends on r, but in Eq. (1) it does not.*
We dropped the dependence on radius because we use the asymptotic value $Q_{ext}=2$ for the drop size regime and retrieval wavelengths considered.

*- Please make sure that regular words are not written in math mode. For example: 'constant' on line 45, top in z_top in Eq. (3).*
All instances corrected

*- L51-52: Physical cross section is usually called geometric cross section.*
Corrected

*- L71: Here r_e becomes a function of z, while before it was a retrieved quantity and not a function of z. Please correct the notation.*
We edited the beginning of this subsubsection to explicitly state that we consider the dependence of aircraft-measured $r_e$ on altitude

*P101: 'A discrepancy ... that agrees ...': Is this sentence correct?*
Revised to: "Witte et al. (2018) found no such bias, with MODIS and in situ measurements agreeing within 0.7 μm in the mean"

*L114: This choice may make sense, but 60 to 90 m below cloud top does not correspond to the middle of the cloud (given that the cloud thickness is between 250 and 500 m, at least for the POST cases, Fig. 10). Please consider changing the term 'mid-cloud'.*
The phrase in question has been revised to:
"We use the range between 60 m to 90 m below cloud top as an analog to level mid-cloud legs to calculate a representative value of $N_d$ during POST. While this altitude range does not correspond to ``mid-cloud" in the sense of cloud geometric thickness, this region is typically far enough from cloud top to avoid the impacts of entrainment mixing."

*L133-136: It would be useful to add some information on the typical time over which the aircraft measurements are aggregated. Given that MODIS sampling is instantaneous, the larger the time window, the larger errors due to temporal variability, including advection, become.*
We now give this information at the end of Section 2.1.2:
"The mean and variability of $N_d$ and $r_e$ for each 10 min (or ≈30 km at a mean true airspeed of 55 m/s) flight leg are calculated from these 1 km average values."

We note that this intercomparison approach has been taken by numerous authors in the past (Painemal and Zuidema 2011, Zheng et al. 2011, Min et al. 2012, Noble and Hudson 2015, Witte et al. 2018). Furthermore, Zheng et al. (2011) found negligible differences applying advective corrections for overpass-in situ measurement time differences of less than about half an hour.

*Eq. 11: This is a measure of uncertainty, but it is not the standard error. Furthermore, there is an implicit assumption that the errors of the individual 1x1 km2 observations are uncorrelated. Is that the case?*

Indeed, this is the 95% margin of error instead of standard error. The text has been updated accordingly, and we also added a sentence afterward to justify our choice of uncertainty variable: "We choose margin of error versus standard error to reflect uncertainty in unsampled spatial variability since the aircraft samples a single transect through a 1x1 km$^2$ box while MODIS senses photons arriving from the entire area."

*Fig. 1: in the axis labels, N should be N_d.*
Figure revised

*Fig. 1: Case 5 (08/14) is not labelled. (Same in Fig. 2. There also the leading 0s are missing from the labels.).*
Figures revised

*L171-174: Can you be more quantitative in what is considered good or bad agreement? A range of +/- 25% is depicted in Figs. 1 and 2, suggesting this is defined as the distinction between good and bad. However, Table 1 contains a case with an r_e difference of 17% which is considered bad agreement.*
Grovsenor et al. (2018) go through each term of the $N_d$ retrieval equation and arrive at an overall uncertainty estimate of ±78%.

*L184-185: In Fig. 1 there are two POST cases for which satellite and in situ observations are well within the +/-25% lines. Why is one of these not considered as good agreement?*
Good catch. We don't know what happened, but we have double-checked all the POST data and they are all accurate. So this second case, which turns out to be on 7/16, is indeed a case with good agreement. We have edited the manuscript to reflect this change. This day is similar to our Case 1 where all underlying variables (profiles of effective radius, beta, and LWC) are in good agreement.

*Section 3.3: For reference it would be useful to include the cloud optical thickness of the POST cases.*
Sawtooth maneuvers do not typically sample the full cloud geometric depth, thus aircraft-derived cloud optical depth is likely underestimated and it is not directly comparable with MODIS retrievals.

*Fig. 4: Liquid water content is referred to as LWC. This should be L, as elsewhere in the paper.*
Corrected as requested in Figs. 4-7

*L207-212: It might be added that, consistent with the underestimate in MODIS cloud optical depth, also the inferred cloud geometrical thickness is too small, i.e. cloud base is too high.*
Done.

*L214: Remove 'a' before 'twice'.*
Done

*L216: PDI does not observe tau_c as such (but it can be inferred from the beta_ext profile), so this sentence is not correct.*

We added the following qualifier to denote that optical depth is not a direct observable:
"...because MODIS and PDI-*derived* tau_c are..."

*L247: 'MODIS assumes ..': rephrase (an instrument does not assume anything).*
Revised to "*The* MODIS $N_d$ retrieval assumes..."

*Fig. 9: Here we seem to be looking at two POST cases that were not discussed before. Where are these located in Figs. 1 and 2? To improve clarity, you could consider to label every POST day (and not just the five of Section 3.3) with a case number (1 to 8), and include a table with the relevant MODIS and PDI cloud properties.*
We have made a number of edits to allow readers to more easily identify the specific POST cases on Figs 1 and 2. We did not include that table, however, as the detailed Figures 4 to 8 give that same information and much more.

*Section 3.4: Would it make sense to be a bit more quantitative about the impact of k on N_d? If k is slightly larger than 0.9, the deviation in N_d is between 10 and 15%. I would agree that this is indeed 'not large' (L251).*
Revised as suggested.

*L274-277: 'MODIS predicts ...': rephrase*
Revised to "the MODIS retrieval predicts"

*L280: Add 'even near the middle of the cloud' (deviations near top/bottom are common as mentioned before)*
We chose not to add this clause as we do not feel it clarifies interpretation of our results.

Reference

Meyer, K., Platnick, S., Arnold, G. T., Amarasinghe, N., Miller, D., Small-Griswold, J., Witte, M., Cairns, B., Gupta, S., McFarquhar, G., and O'Brien, J.: Evaluating spectral cloud effective radius retrievals from the Enhanced MODIS Airborne Simulator (eMAS) during ORACLES, EGUsphere [preprint], https://doi.org/10.5194/egusphere-2024-2021, 2024.

**REVIEWER 4**

The authors assess the accuracy of cloud droplet number concentration ($N_D$) derived from MODIS satellite observations by comparing it to in situ measurements from three field campaigns. Their main finding is that MODIS tends to overestimate $N_D$, with discrepancies of ±50% or more. The authors suggest that these errors stem from variations in microphysical and radiative variables, though no single source of error is identified. However, the results are inconclusive due to the limited dataset, and further high-resolution vertical sampling is needed to establish statistical significance.

Review:

*As a scientist less familiar with this specific topic, I suggest a minor revision. While the study presents important findings, its clarity and structure can be improved, particularly for broader audiences.*

General comments:

*The abstract does not clearly state the main aim or conclusion. Consider adding that this study focuses on evaluating the accuracy of MODIS retrievals using aircraft observations.*

Revised as suggested.

*At the end, include a conclusion stating that more data are needed to better assess the accuracy of MODIS retrievals.*
We elected not to incorporate this suggestion due to our own style preferences.

*$N_D$ is a key focus of the paper, but it is no really clearly defined in the introduction. Briefly define and explain why $N_D$ matters. Similarly, explain more terms like cloud optical depth and cloud effective radius for readers unfamiliar with the field.*

*The introduction is brief and would benefit from:*

o        *A discussion of other methods or campaigns measuring $N_D$ and whether this study is the first of its kind.*

o        *Mentioning other satellite measurements and the importance of accurate $N_D$ values.*

*Including citations to situate the study within the broader research landscape.*

The Introduction was rewritten and incorporates the comments above.

*In section 1.1, several parameters in the equations are not defined immediately and are only explained later in the introduction. This can make it difficult for readers to fully understand the equations when they first encounter them. It would be helpful to define parameters consistently and right when the equations are introduced. For example, Qext is first encountered in line 29, but its definition is provided later, in line 51.*
All parameters in Eq. (1) are now defined immediately following the introduction of the equation.

Specific Line Edits:

- Line 7: *"We find that MODIS N𝒹 is best interpreted as representative of the mid-cloud region, as there is almost always a greater discrepancy from in situ values near cloud top and cloud base " This should be rephrased. Larger differences near the cloud top and base do not necessarily make the mid-cloud region more representative. Please clarify why the mid-cloud region is considered more representative.*
  Discussion of this point was added in the main body of the revised text, see Section 3.3.1 (lines 274-281).

- Line 15: *Correct "cloud drop" to "cloud droplet."*
  The distinction between "drop" and "droplet" is arbitrary and in many ways a relic of simplifications made in bulk parameterizations (i.e., separating "cloud" and "rain" hydrometeor species) and legacy instrumentation (e.g., "droplets" that are observed by one probe and "drops" by another) that is of decreasing relevance. As such, we view the choice of which word to use as a matter of style versus accuracy. We have decided to uniformly employ the term "cloud drop" to remove any ambiguity.

- Line 22: *Rephrase "When we find…" for clarity.*
  The introduction was rewritten and greater context was given before ending with the questions referenced here.

- Line 33: *Instead of indirectly describing the weighting function, state directly: "A weighting function is used to weight the impact of measurements on the satellite-derived variables."*
  Sentence was revised to:
  "To produce an estimate of $r_e$ from aircraft suitable for comparison with that from MODIS, a weighting function is used to weight the impact of cloud vertical structure on the aircraft-derived variables…"

- Line 38: *Add "the" before "cloud top" and provide a citation for the statement about peaking regions. Clarify how cloud optical depth is addressed in this context.*
  References added to support the claim that the weighting function peaks near cloud top.

- Line 42: *Explain that "r" represents droplet radius.*
  Done

- Line 93: *The term "level legs" is unclear. Define it for readers unfamiliar with flight terminology.*
  Sentence was revised to:
  "Flights during the MASE and VOCALS campaigns utilized level legs (i.e., flight segments flown at constant altitude and heading for a sustained period, usually 10 min for the flights analyzed in this work) which sampled from below cloud base to near cloud top."

- Line 106: *Rephrase "for each flight analyzed" to "for each analyzed flight."*
  Done

- Line 111: *Clarify criteria used by flight scientists to select top legs.*
  The phrase in question was removed at the recommendation of another reviewer.

- Line 112: *Reconsider "Therefore" and provide justification for the liquid water content threshold ($L = 0.05$ g/m$^3$).*
  We added the following justification:
  "$L=0.05$ g/m$^3$, a commonly used threshold in the airborne science community"

- Line 117: Define "leg" where it first appears (Line 93).
  Done

- Line 130: Clarify what "Lad(z)" refers to, as only L(z) is defined earlier.
  Revised to: "To determine the adiabatic liquid water content $L_{ad}(z)$"

- Line 192: Explain what is meant by "cloud drop evaporation due to entrainment."
  Revised to: "cloud drop evaporation due to turbulent entrainment of warm, dry air from above the boundary layer"

- Line 236: Change "PDI observation" to "PDI observations."
  Done

- Line 247: Rephrase "ranges $0.7 \leq k \leq 0.9$" to "ranges from 0.7 to 0.9."

  Done